# Eating Disorders in Hospitalized School-Aged Children and Adolescents during the COVID-19 Pandemic: A Cross-Sectional Study of Discharge Records in Developmental Ages in Italy

**DOI:** 10.3390/ijerph192012988

**Published:** 2022-10-11

**Authors:** Gianmarco Giacomini, Heba Safwat Mhmoued Abdo Elhadidy, Giovanni Paladini, Roberta Onorati, Elena Sciurpa, Maria Michela Gianino, Alberto Borraccino

**Affiliations:** 1Department of Public Health Sciences and Pediatrics, Università di Torino, 10126 Torino, Italy; 2Regional Public Health Observatory, Epidemiology Unit, Local Health Board TO3, Piedmont Region, 10195 Grugliasco, Italy

**Keywords:** eating disorders, hospitalization, trend, COVID-19, mental health, developmental age, healthcare

## Abstract

Eating disorders (EDs) are characterized by behavioral and cognitive aspects that result in a significant impairment of an individual’s well-being. COVID-19 pandemic consequences negatively impacted healthcare services and people’s mental health. Particularly, in developmental ages, difficulties in coping with the situation could have had an impact on eating behaviors. Therefore, the aim of this study was to assess EDs’ hospitalization trend before, during and after the pandemic peak to evaluate whether it has been influenced. A retrospective cross-sectional study was conducted on the hospital discharge forms of patients from 5 to 19 years old in Piedmont, which is a region in northern Italy. Overall, hospitalization, age, and gender-specific rates due to EDs that occurred in 2020 and 2021 were compared to those that occurred in 2018–2019. Since 2020, there has been a 55% reduction in overall hospitalizations, while the total proportion of EDs admissions has doubled from 2020 to 2021 (from 13.9‰ to 22.2‰). Significant hospitalization rate increases were observed both in 15–19 and in 10–14 females’ age groups in 2021. Non-significant increases were observed in all males’ age groups. The increase in hospitalizations for EDs should be further investigated, as it might be the tip of an iceberg not yet acknowledged.

## 1. Introduction

Eating disorders (EDs) are described in the fifth edition of the Diagnostic and Statistical Manual of Mental Disorders (DSM-5) as “a persistent disturbance of eating or eating-related behavior that results in the altered consumption or absorption of food and that significantly impairs physical health or psychosocial functioning” [1]. The heterogeneous data and the complexity of diagnosis make it difficult to assess EDs’ global prevalence, but it can be stated that EDs are highly prevalent worldwide and seem to be increasing. The mean prevalence went from 3.5% in the 2000–2006 period to 7.8% in the 2013–2018 period [2]. According to available pre-pandemic data in Italy, EDs’ prevalence was 0.2–0.8% for anorexia nervosa and 1–5% for bulimia nervosa [3,4]. EDs affect both genders, even if they are more diagnosed in women. They usually emerge during teenage years, with genetic, biological, behavioral, psychological, and social factors involved in the pathogenesis [5,6]. Among these aspects, the individual’s and the family’s high socioeconomic and educational status seem to have a great influence on EDs’ development process [7]. EDs expose individuals to many psychiatric and somatic complications, ranging up to being the most life-threatening psychiatric disorders in the US [5,6]. Over 70% of individuals with EDs report comorbidities such as malnutrition, anxiety, mood disorders, depression, self-harm, and substance abuse. In this respect, they lower patients’ quality of life and increase mortality, with an elevated risk of suicide, which is higher for anorexia nervosa [2,8]. Gauging the burden and determinants of EDs will be fundamental in order to mobilize policy makers. It seems that even before the pandemic only a portion—approximately one-third—of individuals with an ED was detected and treated, so that the prevalence we know could be only the tip of the iceberg [8].

The COVID-19 pandemic and the related restrictions to social life, enacted from March 2020 and persisting in varying degrees until today, heavily impacted individuals, especially the vulnerable ones. Indeed, previous literature reported a significantly poorer mental health status in those who suffered from EDs than the general population during the pandemic [9]. Social isolation, anxiety around family matters and economic factors [9], as well as a perceived low quality of therapeutic relationship and fear of contagion all negatively acted upon their mental well-being [10]. Seeking and receiving the adequate treatments was tougher as a consequence of the reallocation of resources and the disruption of routinary healthcare [11,12,13]. Moreover, children and adolescents were particularly affected by social restrictions, as important pillars of their social and educational routines were disrupted, bringing them to isolation and inactivity [11,14]. Hence, it is reasonable to assume an increase in ED cases due to this unprecedented context: this happened in other countries, as stated by different authors [11,14], and it was expected to happen in Italy as well. Official data were not retrieved, and information was only spread anecdotally by news outlets (that reported a 30% rise in new cases [15,16]).

On top of that, the lack of research on younger ages is vast, especially considering that in those ages, the foundations for people’s mental well-being are formed. In fact, those are very delicate ages in which personality and identity are developed. It is mostly during adolescence that the youth learn more about themselves, eminently thanks to certain life experiences and transitions (such as having the first romantic relationship, leaving the parental home or getting the first job) that stimulate personality maturation [17]. This aspect is particularly important when speaking of conditions that can occur early in life and can intensify as time goes by, such as EDs. Indeed, it is believed that there is a continuum from normal eating to full ED that could evolve over time [18]; therefore, the goal would be to intervene at an early stage of this process rather than focusing on the already developed disorder.

To authors’ knowledge, there is still no study in Italy evaluating the EDs hospitalization trend in school-aged children and adolescents and the influence of COVID-19 on it. Therefore, the aim of this study was to assess EDs’ hospitalization trend over the last four years (2018–2021) in Piedmont, a northern Italian region, in order to detect changes due to the pandemic, stratifying by gender, age, diagnosis and origin of hospitalization

## 2. Materials and Methods

### 2.1. Study Design, Setting and Participants

*A* retrospective cross-sectional observation of all hospital admissions that occurred in the Piedmont region was performed over three intervals of time: the pre-pandemic period (2018–2019), during the COVID-19 pandemic (2020), and the year after the pandemic peak (2021). The year 2018 and the year 2019 served as a baseline to define the pre-pandemic period. Piedmont is the second largest region in Italy, with a population of more than 4.3 million inhabitants over an area of more than 25,300 km^2^. Approximately 500,000 of its residents are school-aged, between 5 and 19 years old.

Information on hospital admissions that occurred in any regional hospital between January 2018 and September 2021 (last trimester was not available due to administrative reasons) were retrieved through the official Italian National Information Discharge System database. Data included demographic information (gender and date of birth), date of admission and discharge, diagnosis coded according to the International Classification of Diseases, ninth Revision, Clinical Modification (ICD9-CM) [19], and origin of hospitalization coded as being an Emergency Department access, a planned admission, a transfer from another structure or another department, or a direct dispatch by a pediatrician or a general practitioner (for adolescents between 15 and 19 years old that are no longer followed by a pediatrician). Hospital discharge reporting a primary or secondary diagnosis of an ED included anorexia nervosa (307.1) and other/unspecified EDs (codes between 307.50 and 307.59), which comprise bulimia nervosa, binge eating, rumination disorders and pica.

### 2.2. Ethics

Data were obtained by accessing the official Italian National Information Discharge System database through the universal anonymous patient identification number. The number is a certified univocal non-reversible anonymous code, which was centrally assigned before data storage. The code allows data management to previously accredited institutions, without any further authorization. As all administrative ministerial data are made available to authorized bodies in a fully anonymized and de-identified manner, an Ethics Committee approval was not required.

### 2.3. Statistical Analyses

All hospital discharges of children and adolescents aged 5 to 19 years old were retrieved for each year in study, between 2018 and 2021. Then, the investigation focused on medical charts for patients who, coherently with the diagnostic criteria, had been hospitalized for an ED, independently of whether it represented a first-time hospitalization or a re-admission. Target patients were grouped by gender in three age categories: 5–9, 10–14 and 15–19 years old.

Descriptive analyses were reported for each year between 2018 and 2021, stratified by gender and age groups. Age-specific crude rates, and 95% confidence intervals (CIs), were calculated based on the resident population of the Piedmont Region in each year and expressed as number of cases per 100,000 inhabitants. To account for differences in the age distribution, direct standardization with 95% CIs was also calculated to report the overall rate of EDs (between 5 and 19 years old) by using 2019 data for the general population of the Piedmont region. To account for any possible variations in hospital admissions, the ratio of EDs for each year of observation, per thousand of EDs admissions with a 95% CI, was also considered. Analyses were performed using SAS version 9.4 (SAS Institute Inc., Cary, NC, USA).

## 3. Results

EDs admissions were assessed as the ratio of EDs admissions per 1000 overall hospitalizations and the rate of EDs admissions per 100,000 overall hospitalizations in children and teenagers aged from 5 to 19 years old, divided into 5-year categories (5–9; 10–14; 15–19). The ratio and the rate above-mentioned were assessed before (2018–2019), during (2020) and after (2021) the pandemic peak.

The analysis carried out indicates that both hospitalizations for all causes and ED-specific hospitalizations were stable up to 2019, as shown in Figure 1. Since then, up to September 2021, the number of overall hospitalizations for any cause in developmental ages was reduced by 55%, while the EDs’ admissions ratio per 1000 overall admissions increased significantly from 11.4 in 2019 to 22.2 in 2021. In the same period, ED-related first-time hospitalizations increased from 62.2% in 2018 up to 74.18% in 2021 and, conversely, the number of subsequent hospitalizations decreased.

According to the ratio assessment (shown in Table 1), in the female group, the 15–19 age range produced a significant increase in hospitalizations from 40‰ (CI 34.5–47.8) in 2020 to 67.8‰ (CI 58.1–79.0) in 2021. Instead, considering the age-adjusted ratio from 5 to 19 years old, a significant increase was detected from 24.76‰ in 2020 (CI 21.4–28.1) to 40.57‰ in 2021 (CI 35.3–45.8). As for the male group, no significant changes were detected, with increases in all age ranges, but these changes were not statistically significant. 

Moreover, the rate assessment yielded significant raises (shown in Table 2). In the female group, the 10–14 age range showed a significant increase from 57.69 (CI 44.2–75.3) EDs admissions per 100,000 inhabitants in 2020 to 117.08 (CI 91.7–149.5) EDs admissions in 2021. The 15–19 age range showed a significant increase from 149.95 (CI 127–177) EDs admissions per 100,000 inhabitants in 2020 to 282.65 (CI 241.2–331.2) EDs admissions in 2021. Considering the age-adjusted rate from 5 to 19 years old, a significant increase in EDs admissions per 100,000 inhabitants was detected from 70.8 (CI 61–80.7) EDs admissions in 2020 to 135.5 (CI 117.6–153.4) EDs admissions in 2021. As for the male group, no significant changes were detected, even if in all age ranges, there were non-significant increases, especially a 2021 peak in the 10–14 age group.

No changes about the diagnosis trend have been detected in the last four years: anorexia nervosa was the most diagnosed disorder (55.3%), followed by unspecified eating disorders (34.8%), then by bulimia nervosa (5.7%); data are not shown in tables.

The healthcare pathway of patients with EDs was evaluated in order to precisely identify which healthcare professional suggested the hospitalization. EDs admissions’ proportion of all hospitalization events by origin of hospitalization (shown in Table 3) yielded a significant increase from 38.9‱ (CI 30.6–49.5) in 2020 to 93.3‱ (CI 77.1–112.8) in 2021 in the number of patients coming from emergency departments as well as a significant increase from 13.8‱ (CI 9.9–19.2) in 2019 to 28.3‱ (CI 21.4–37.5) in 2020 in the number of patients sent by pediatricians; these data kept increasing in 2021, even if the rise was not significant. Moreover, there was a significant increase in the number of patients coming from other wards or services, from 17.7‱ (12.4–25.3) in 2020 to 34.6‱ (25.3–47.3) in 2021.

## 4. Discussion

This study assessed EDs’ hospitalization trends during the last four years and how the COVID-19 pandemic acted upon it. Hospital admission trends for all causes declined starting from 2020: both routine and emergency care were reduced because of the pandemic. This general drop was noted worldwide and not only in Italy [20,21]: for example, in the US, the hospitalization trend for non-COVID related illness decreased by 21% [22]. By contrast, the results of this study indicate that EDs’ hospitalization trends have remained stable or increased since the pandemic compared to hospitalizations due to other causes (Figure 1). In particular, it was observed that first-time admissions for ED had increased from 2018 to September 2021, with a concomitant reduction in overall readmissions.

As for the EDs’ diagnosis trend in the hospitalized population, the findings of this study showed that during the studied years, anorexia nervosa was still the most common diagnosis, which was followed by unspecified eating disorders. The latter diagnosis contains all those subclinical or atypical symptoms not meeting full criteria for any other ED. Despite being one of the most common diagnoses of ED, it has not been included in the Global Burden of Disease study until 2019, and it would need further assessment in order to be fully comprehended [23].

During 2020, there has been an increased need for ED-related care, which doubled in 2021. This increase was already being detected in the previous years: the Global Burden of Disease study, carried out in 2019, estimated that 2.91 million (95% UI 1.83–4.35) global DALYs were lost due to EDs in 2019, with a 14% increase in EDs’ prevalence and a 13.7% DALYs increase for both sexes since 2010 [23,24]. However, numerous studies suggest an association between the COVID-19 pandemic and EDs’ development or exacerbation in youth, thus prompting more medical admissions and rapid readmissions among school-aged children and adolescents [25]. A prominent aspect, evaluating the effects of the pandemic on EDs, lies within the reactivation of EDs’ symptoms during confinement [26]. Some examples are a decrease in eating frequency and quantity, excessive exercising, fear of gaining weight and increased emotional symptoms. Trying to estimate the magnitude of the combined increase in EDs’ trends reported by the scientific community in the past two years is extremely difficult, but the results are alarming. 

Stratifying EDs hospitalizations by gender and age, significant increases have been found. The 15–19 years old females’ increase is noteworthy: EDs are known to be prevalent at age 15–39 [27], and nearly all first-time cases of EDs occur by 25 years of age (both in females and males) [28]. As for the age of onset, there are discordant findings in the literature: according to some researchers [29], the peak age for anorexia nervosa diagnosis would be between 11 and 14 years old, whereas diagnosis of bulimia nervosa would be more frequent between 15 and 19 years old. Instead, as stated by others [30], a mean age of onset of about 18 years was seen both for anorexia nervosa and bulimia nervosa. These discrepancies could be due to numerous reasons. Firstly, the lack of an unambiguous definition of the onset of EDs could lead to a delay in diagnosis or to underdiagnosis, especially in a primary care context [31]. Secondly, contrary to the diagnostic instruments currently in use, questionnaires with flexible cut-off points would be indispensable to identify sub-threshold illness. Furthermore, many frequently used tools are designed to identify traditional presentations of anorexia nervosa and bulimia nervosa but are not capable of recognizing unspecified eating disorders [32]. Moreover, the ‘first-time’ diagnosis of the disorder could not be the same as the true age of onset [30,33]. Lastly, the first symptom is often a very common behavior (e.g., caloric restraint or food avoidance), which is difficult to distinguish from a non-pathological one. Therefore, it is worth distinguishing between the onset of symptoms and the age of first diagnosis, since the two events could be distant in time. The literature underlines how the age of onset has been decreasing [34]. An untreated earlier age of onset is a negative prognostic factor in terms of response to treatment; however, it does not appear to influence rehospitalizations [35,36]. 

Increases in males’ hospitalization ratio were found in all age groups, even if they were non-significant. Usually, it is more common for males to be diagnosed or to need hospitalization before the age of 15, even earlier than females [36,37,38,39]. After childhood, hospitalization seems to be less needed by males for several reasons. Firstly, they are susceptible to the social stigma associated with mental health treatment. In fact, traditional social expectations on males, such as independence, resilience, and self-reliance, inhibit help-seeking behavior. Secondly, the stigma attached to EDs, being considered a “women’s disease”, exacerbates the problem [40]. Lastly, males’ reluctance to seek help depends on how EDs symptoms are perceived and minimized, either by themselves or by society. For example, excessive exercising goes hand in hand with the muscular idealization of the male figure. In this respect, it is important to consider that males with EDs might be interested in gaining weight as well as in losing it [36,38]. A rise in males’ hospitalizations for EDs after 2020 is probably linked to COVID-19 pandemic [41], making it important to monitor young adult males in times to come.

Transition to adulthood adds a challenging dynamic for both sexes in Italy: outpatient clinicians and primary healthcare programs require a deliberate enrollment in treatment. In addition, Italian youngsters are followed from zero to 14 years of age by pediatricians, while from 14 years onwards, they must rely on general practitioners [42]. This transition process, if managed inappropriately, could lead to a decrease in the continuity of care, and some patients may be lost to follow-up. By contrast, it can be successful if preceded by an adequate preparation and discussion on the most effective treatment plan. When preparing youth for adulthood, it is important that clinicians make sure that all the figures involved are willing to follow the recommendations agreed upon. Parents, on their side, need to provide emotional and economic support to encourage young adults to participate in the necessary treatment [43]. On the other hand, if emerging adults do not receive adequate care, they could be reluctant to seek help on their own. This seems to be related to the shame and embarrassment felt by patients in talking about their disorder. There is evidence that believing to be too old for EDs may be a particularly pertinent barrier to seeking help. Dietary control and weight loss are perceived to boost confidence and help cope with stressful situations [44].

Taking into account the provenance of the hospitalizations, a change has been noted for EDs. A significant rise was found in hospitalized patients coming from the emergency department. This is consistent with other studies across the world, especially papers from Australia [45], the UK [46] and the US [25,47]. Moreover, in Canada [48], researchers observed how the patients that accessed the emergency department during the COVID-19 pandemic were more medically unstable and functionally impaired compared to those who accessed it in 2019. These changes may have been due to several factors. Firstly, the pandemic acted either as a trigger for the acute onset of the illness or as the reason for the closure of some primary care clinics. In many countries, the worsening of EDs’ routine healthcare made it more difficult to access proper treatments and symptoms’ surveillance [12,25,49] and could have led patients to visit the emergency department more frequently. This is an event that could have possibly occurred in Italy too, so that close attention will be needed [50]. Secondly, lockdown could have played a role, since parents may have been able to notice earlier changes in their children’s eating behavior or other alarming behaviors, being confined in the house together for days. In addition, this study displayed a significant increase in patients sent by family pediatricians. It is already known how general practitioners and pediatricians feel uncertain about the best course of action with patients suffering from EDs [51,52]. It is possible that working under different conditions during the COVID-19 pandemic—such as practicing telemedicine and with less chance to refer patients to specialist care—could have grown that feeling of uncertainty. For example, many health professionals used telemedicine for the first time during the pandemic, and, not yet accustomed to it, they reported a certain discomfort and a lack of concentration, both for themselves and their patients. This background, along with a compromised nonverbal communication and the lack of physical evaluation, complicated the therapeutic alliance and medical monitoring of patients with EDs [50,51]. Moreover, a more severe manifestation of symptoms or a better recognizability of non-diagnosed EDs could have led to a rise in patients sent by pediatricians to the emergency department [41].

In conclusion, the ratio of patients transferred from other wards or services due to EDs increased significantly in 2021. Transferring could have many drawbacks, such as the discontinuation of medications or even minor or severe adverse events [53]. The pandemic made this process more complicated because of the necessary precautions [54], but it is still indispensable, especially for patients with a delicate clinical stability, such as those with an ED.

### Limitations

This study has some limitations. Major limitations are those regarding the information sources used and are common to all administrative database studies. Collected data were extracted from hospital discharge forms. The quality of information is operator-dependent, so the coding criteria used by individuals and across different institutions can vary for precision or accuracy. A further limitation concerns the availability of data that was restricted to September 2021. It is likely that despite the observed increase in the proportion of hospitalizations, the trend may have continued, so further studies are needed. Finally, as the aim of this study was to assess the volume of admissions for ED in the Piedmont region, it is not possible to deduce the real incidence of ED in the pre-adolescent and adolescent population, which could potentially be overestimated. However, it has to be noted that the substantial growth in first-time hospitalizations could suggest a possible increase in new EDs, which may deserve further investigation.

## 5. Conclusions

This study points out the importance of analyzing EDs’ admissions trend in order to understand whether the of EDs phenomenon is changing. It appears that over the last few years, there has been an increased healthcare need among specific populations; further studies will be fundamental to fully comprehend this process and to follow it over time. While drawing attention to the fact that EDs seem to be a non-deferrable emergency, our findings suggest that this growing trend should not be overlooked, as it may get worse.

It would be important to improve prevention and primary health care concerning EDs, such as advocating for multidisciplinary teams among health professionals (specialists, as well as family doctors and family nurse practitioners), working together from prevention to follow-up [51]. Telemedicine must not be underestimated, as it could guarantee a close and more frequent engagement of some specific group of patients in time to act upon behaviors before the pathology’s onset. 

To completely understand how the COVID-19 pandemic impacted on routinary activities, it is important to consider that EDs could not be the only area of interest revealing a changed trend.

## Figures and Tables

**Figure 1 ijerph-19-12988-f001:**
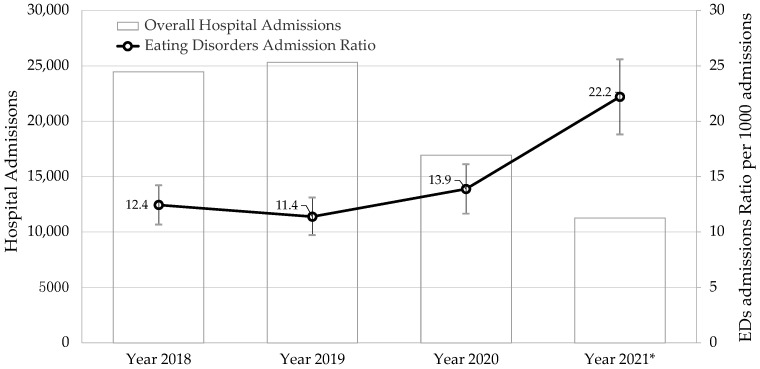
Overall hospital admissions and age adjusted (with 95% confidence intervals) EDs ratio of hospital admissions per 1000 admissions, pre (year 2018 and 2019), during (2020) and post (2021) pandemic peak; Piedmont region, Italy (* Year 2021 comprises data from 1 January 2021 to 30 September 2021 due to dataset information availability).

**Table 1 ijerph-19-12988-t001:** Gender age-specific and age-adjusted (with 95% CIs) EDs ratio of hospital admissions per 1000 admissions, pre (years 2018 and 2019), during (2020) and post (2021) the pandemic peak; Piedmont region, Italy.

		Year 2018 ^‡^			Year 2019 ^‡^			Year 2020 ^‡^			Year 2021 ^‡,§^		
		N	Ratio	(95%CI)	N	Ratio	(95%CI)	N	Ratio	(95%CI)	N	Ratio	(95%CI)
Females		10,963			11,303			7700			5194		
	5 to 9	4	1.23	(0.5–3.2)	8	2.47	(1.3–4.9)	4	1.94	(0.8–5.0)	4	3.17	(1.2–8.1)
	10 to 14	49	16.54	(12.5–21.8)	49	15.24	(11.5–20.1)	54	24.29	(18.7–31.6)	64	37.87	(29.8–48.1)
	15 to 19	196	41.24	(35.9–47.3)	175	36.13	(31.2–41.8)	139	40.64	(34.5–47.8)	152	67.80	(58.1–79.0)
	5 to 19 *	249	22.71	(19.9–25.5)	232	20.51	(17.9–23.1)	197	24.76	(21.4–28.1)	220	40.57	(35.3–45.8)
Males		13,500			14,028			9245			6066		
	5 to 9	5	1.17	(0.5–2.7)	4	0.91	(0.4–2.3)	2	0.73	(0.2–2.6)	0	-	-
	10 to 14	10	2.41	(1.3–4.4)	9	2.07	(1.1–3.9)	9	2.95	(1.6–5.6)	13	6.33	(3.7–10.8)
	15 to 19	14	2.76	(1.6–4.6)	19	3.58	(2.3–5.6)	17	4.95	(3.1–7.9)	11	5.06	(2.8–9.0)
	5 to 19 *	29	2.15	(1.4–2.9)	32	2.27	(1.5–3.1)	28	3.00	(1.9–4.1)	24	3.85	(2.3–5.4)

* age-adjusted ratio; ^§^ Year 2021 comprises data from 1 January 2021 to 30 September 2021 due to dataset information availability ^‡^ First-time hospitalization, total male and female in 2018: 62.2; 2019: 62.1%; 2020: 67.6%; 2021: 74.2%.

**Table 2 ijerph-19-12988-t002:** Gender age-specific and age-adjusted (with 95% CIs) EDs hospital admissions rate per 100,000 inhabitants, pre (2018–19), during (2020) and post (2021) the pandemic peak; Piedmont region, Italy.

		Year 2018		Year 2019		Year 2020		Year 2021 ^§^	
		Rate	(95%CI)	Rate	(95%CI)	Rate	(95%CI)	Rate	(95%CI)
Females	5 to 9	4.34	(1.7–11.2)	8.95	(4.5–17.7)	4.59	(1.8–11.8)	8.04	(3.1–20.7)
	10 to 14	52.40	(39.6–69.3)	52.61	(39.8–69.5)	57.69	(44.2–75.3)	117.08	(91.7–149.5)
	15 to 19	211.39	(183.8–243.1)	189.46	(163.4–219.7)	149.95	(127.0–177.0)	282.65	(241.2–331.2)
	5 to 19 *	89.44	(78.3–100.5)	84.41	(73.0–94.5)	70.81	(61.0–80.7)	135.56	(117.6–153.4)
Males	5 to 9	5.08	(2.2–11.9)	4.20	(1.6–10.8)	2.16	(0.6–7.9)	-	-
	10 to 14	10.04	(5.5–18.5)	9.07	(4.8–17.2)	9.05	(4.8–17.2)	22.40	(13.1–38.3)
	15 to 19	13.93	(8.3–23.4)	19.13	(12.3–30.0)	17.17	(10.7–27.5)	19.20	(10.7–34.4)
	5 to 19 *	9.71	(6.2–13.3)	10.89	(7.1–14.6)	9.58	(6.0–13.0)	13.89	(8.3–19.3)

* age-adjusted ratio; ^§^ Year 2021 data comprises data from 1 January 2021 to 30 September 2021 due to dataset information availability.

**Table 3 ijerph-19-12988-t003:** Origin of hospitalization for EDs admission, absolute frequency, and ratio (with 95% CIs) per 10,000 admissions; pre (years 2018 and 2019), during (2020) and post (2021) the pandemic peak; Piedmont region, Italy.

	Year 2018			Year 2019			Year 2020			Year 2021 ^§^		
	N	Ratio	(95%CI)	N	Ratio	(95%CI)	N	Ratio	(95%CI)	N	Ratio	(95%CI)
Em. Dept. Access	52	21.3	(16.2–27.9)	80	31.6	(25.4–39.3)	66	38.9	(30.6–49.5)	105	93.3	(77.1–112.8)
By pediatrician	35	14.3	(10.3–19.9)	35	13.8	(9.9–19.2)	48	28.3	(21.4–37.5)	44	39.1	(29.1–52.4)
Planned Admission	98	40.1	(32.9–48.8)	80	31.6	(25.4–39.3)	72	42.5	(33.8–53.5)	46	40.9	(30.6–54.4)
Transferred	53	21.7	(16.6–28.3)	45	17.8	(13.3–23.8)	30	17.7	(12.4–25.3)	39	34.6	(25.3–47.3)
Unspecified	40	16.4	(12.0–22.3)	24	9.5	(6.4–14.1)	9	5.3	(2.8–10.1)	10	8.9	(4.8–16.3)

Em. Dept.—Emergency Department; ^§^ Year 2021 comprises data from 1 January 2021 to 30 September 2021 due to dataset information availability.

## Data Availability

The datasets generated and/or analyzed during the current study are not publicly available as data are not public, but they are available from the corresponding author upon reasonable request.

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
