# Peer review of "Eating Disorders in Hospitalized School-Aged Children and Adolescents during the COVID-19 Pandemic: A Cross-Sectional Study of Discharge Records in Developmental Ages in Italy"

_ijerph, 2022, doi:10.3390/ijerph192012988_

Round 1

Reviewer 1 Report

The paper implies eating disorders have increased over the pandemic but the evidence presented does not support this. If you look at the actual number of cases, they have decreased over time.  For females from 249 in 2018 to 220 in 2021 and for males from 29 in 2018 to 24 in 2021.  Although these cases have become a higher proportion of the admitted cases, this is most likely because treatment for serious EDs cannot be delayed. 

There were significant decreases in the total number of hospital admissions and in the conclusion the authors state that EDs seem to be a non-deferrable emergency.  This is really the main message from this article.   

Author Response

Reviewer #1

Open Review

Comments and Suggestions for Authors

The paper implies eating disorders have increased over the pandemic but the evidence presented does not support this. If you look at the actual number of cases, they have decreased over time.  For females from 249 in 2018 to 220 in 2021 and for males from 29 in 2018 to 24 in 2021.  Although these cases have become a higher proportion of the admitted cases, this is most likely because treatment for serious EDs cannot be delayed. 

Authors’ reply (AR): Dear Reviewer, We thank you for the remarks and we agree with your observation: as you rightly report, the absolute number of cases has decreased over time, as has the number of all-cause hospitalizations. Besides, shifting the focus to the gender age specific and age adjusted EDs hospital admissions ratio (Table 1), it can be noticed an increase both in female and in male (not statistically significant in this last case). Therefore, proportionally, more children and adolescents’ patients are hospitalized because of EDs. Among all possible explanations for this phenomenon it appears that treatment for serious EDs is perceived as a not deferable condition.

There were significant decreases in the total number of hospital admissions and in the conclusion the authors state that EDs seem to be a non-deferrable emergency.  This is really the main message from this article.  

AR: We agree with your statement and we thank your for the remark, this is one of the main messages of our study, and it was reported in the first paragraph of the discussion.

Reviewer 2 Report

I appreciate the opportunity to review this article entitled Eating disorders in the adolescent population during the COVID-19 pandemic: a cross-sectional study of discharge records in developmental ages in Italy. Below are some recommendations for improvement.

1.- Title. Consider whether it would be appropriate to refer to the fact that the sample is referred to hospitalized patients.

2.- Line 52. Expressly indicate that it refers to the Covid pandemic, and indicate the time period.

3.- Line 77. Indicate the specific years.

4.- Line 83. Indicate the specific years of the three periods.

5.- Line 108. Explain why research supervision by an ethics committee is not necessary.

6.- Line 116. According to the title, the investigation is in adolescents, but 5-year-old children have been investigated. The title and the comments of the introduction, objectives, discussion... about the fact that the sample is only of adolescents should be adjusted.

7.- Consider including some more trend figures, for example one that shows the evolution and comparison of the age groups.

Author Response

Reviewer 2

Open Review

Comments and Suggestions for Authors

I appreciate the opportunity to review this article entitled Eating disorders in the adolescent population during the COVID-19 pandemic: a cross-sectional study of discharge records in developmental ages in Italy. Below are some recommendations for improvement.

Authors’ reply (AR): Dear Reviewer, we are extremely pleased you have appreciated our study. As you will notice, we have taken into consideration all your recommendation.

1.- Title. Consider whether it would be appropriate to refer to the fact that the sample is referred to hospitalized patients.

AR: Thank you for your suggestion: the proposed adjustments have been included.

“Eating disorders in hospitalized school-aged children and  adolescents hospitalized during the COVID-19 pandemic: a cross-sectional study of discharge records in developmental ages in Italy”

2.- Line 52. Expressly indicate that it refers to the Covid pandemic, and indicate the time period.

AR: The proposed adjustments have been included.

“The COVID-19 pandemic and the related restrictions to social life, enacted from march 2020 and persisting in varying degrees until today in relation to the severity of the peaks, heavily impacted individuals, especially the vulnerable ones.“

3.- Line 77. Indicate the specific years.

AR: The proposed adjustments have been included.

Therefore, the aim of this study was to assess EDs’ hospitalization trend over the last four years (2018-2021) in  Piedmont, a northern Italian region, in order to detect changes due to the pandemic, stratifying by gender, age, diagnosis and origin of hospitalization.”

4.- Line 83. Indicate the specific years of the three periods.

AR: The proposed adjustments have been included.

“A retrospective cross-sectional observation of all hospital admissions that occurred in the Piedmont Region was performed over three intervals of time: the pre-pandemic period (2018-2019), during the COVID-19 pandemic (2020), and the year after the pandemic peak (2021).”

5.- Line 108. Explain why research supervision by an ethics committee is not necessary.

AR: As reported in the paragraph “Ethics”, for this study we have used only administrative ministerial data. This kind of data is made available only to authorized bodies in a fully anonymized and de-identified manner: for that reason, according to ministerial agreements, no Ethics Committee approval is required. However, the same issue has already been discussed with the Editor. Hence, as agreed with the editor, there is no to further specify it along the manuscript.

6.- Line 116. According to the title, the investigation is in adolescents, but 5-year-old children have been investigated. The title and the comments of the introduction, objectives, discussion... about the fact that the sample is only of adolescents should be adjusted.

AR: The proposed adjustments have been included: the term “adolescent” was changed to “school-aged children and adolescents” both in the title and in the body of the text.

7.- Consider including some more trend figures, for example one that shows the evolution and comparison of the age groups.

AR: Thank you for this comment. We agree numbers can be cumbersome for a part of the public. Therefore, we discussed extensively the possibility of adding other figures, as suggested. On the same time, given that the main concepts were already reported in the tables and no new information would have been added, to avoid redundancies we have agreed to not include any other figure.

Reviewer 3 Report

The paper reports an interesting study about the effects of the COVID-19 pandemic on adolescent hospitalization. This is a relevant topic all over the world, with both clinical and management implications. Please, find below my comments for the authors:

- abstract: more than behaviors, EDs are psychiatric conditions characterized by both behaviors and cognitive aspects. Please, revise the first sentence.

- lines 31-34: as in the abstract, EDs are not limited to behaviors or food choices (otherwise, you should write only about ARFID). Please, be more careful in describing the disorders.

- lines 38-39: Italian epidemiologic data reports different data than the one quoted by the authors. Please, evaluate this aspect: see  Favaro et al., 2003 doi: 10.1097/01.PSY.0000073871.67679.D8

- line 40: be aware that EDs are underdiagnosis in males, so I would not say that they are more frequent in women (see https://doi.org/10.1080/10640266.2012.715512), maybe you might say that they are more diagnosed in women. 

- please include quotes for lines 52-55. 

- quotes 11 and 12: this is a scientific paper, you should quote scientific data and not newspapers. I think the possible effects of the COVID-19 pandemic on ED should be described here better. Previous studies have reported specific vulnerability factors linked to the exacerbation of the ED symptoms in patients (https://doi.org/10.1007/s40519-020-01097-x), as well as discussed the incidents (https://doi.org/10.1192/bjp.2021.105), also with longitudinal evaluations (https://doi.org/10.1016/j.eatbeh.2021.101564)

- It is not clear to me if second or third admission might be detected and excluded from the analysis.

- for 2021, a quarter of the data is missing. This is a serious limitation of the study! and should be stated clearly in the limits. Is it possible to delete the same quarter in all the years and evaluate the effects? 

-  line 166: why you did not report data for BN?

- lines 187-191: please be aware that your data is in line with other studies showing an increase in pediatric admission for EDs from the end of 2019, especially with the COVID pandemic (see http://dx.doi.org/10.1136/archdischild-2020-319868). Please, revise the paragraph. 

- line 208: it is not clear if the authors are referring to EDNOS or to AN as the most prevalent ED. In any case, it is not true; BED diagnosis has a higher prevalence over the other diagnosis in the general population (see 10.1097/yco.0000000000000449). Maybe, it is true that AN is the most common diagnosis for hospitalization, and it is due to its complication. Moreover, quotes #19 is uncorrected

- Overall, the discussion is too speculative. The authors should be more closed to their data while they discussed the Italian national health system, the definition of onset for ED, the role of pediatricians and the shame felt by parents. These are all aspects far from your data and far from what you can say with your data (and quotes are scarce). 

- please include in your limits the retrospective nature of the study, as well as that your data are limited to the public hospital (I think), and maybe you lost patients that were directly hospitalized in private facilities for ED treatments (maybe this could be more effective for waves 2018-2019 when EDs were less treated in general hospital?).

- I think the paper needs a revision by a Native speaker. Some paragraphs are quite hard to read.

Author Response

Reviewer 3

Comments and Suggestions for Authors

The paper reports an interesting study about the effects of the COVID-19 pandemic on adolescent hospitalization. This is a relevant topic all over the world, with both clinical and management implications. Please, find below my comments for the authors:

Authors’ reply (AR): Dear Reviewer, we are pleased you have appreciated our manuscript. We’d like to thank you for your comments, which were essential to improve our article.

 - abstract: more than behaviors, EDs are psychiatric conditions characterized by both behaviors and cognitive aspects. Please, revise the first sentence.

AR: We agree with your consideration. The first sentence of the abstract has been revised accordingly.

Eating disorders (EDs) are characterized by behavioral and cognitive aspects that result in a significant impairment of an individual’s well-being.”

- lines 31-34: as in the abstract, EDs are not limited to behaviors or food choices (otherwise, you should write only about ARFID). Please, be more careful in describing the disorders.

AR: Thank you for your suggestion. To be more accurate we have decided to follow and report the official definition from DSM-5th edition: “EDs are persistent disturbance of eating or eating-related behavior that results in the altered consumption or absorption of food and that significantly impairs physical health or psychosocial functioning”

- lines 38-39: Italian epidemiologic data reports different data than the one quoted by the authors. Please, evaluate this aspect: see  Favaro et al., 2003 doi:

AR: Thank you for the suggestion and the references. We would bring the attention to the the fact that data reported in the mentioned study dates back to nearly 20 years ago. Selected quotes refers to more up to date prevalence data, which were further estimated by the Italian National Institute of Health. Nevertheless, when comparing the data we reported in our study (0.2-0.8% for anorexia nervosa and 1-5% for bulimia nervosa) with the data mentioned in the article suggested (0.3% for anorexia nervosa and 1.8 % for bulimia) we’ve noticed that they are highly comparable (regarding the current prevalence).

- line 40: be aware that EDs are underdiagnosis in males, so I would not say that they are more frequent in women (see https://doi.org/10.1080/10640266.2012.715512), maybe you might say that they are more diagnosed in women. 

AR: Thank you for this comment: we agree and the proposed adjustments have been included.

EDs affect both genders, even if they are more diagnosed in women.”

- please include quotes for lines 52-55. 

AR: We would like to point out that the quotations in the following few lines refer to the entire previous period. Besides, to account for the reviewer’s indication we have amended the sentences syntax to be more accurate with the quotes.

- quotes 11 and 12: this is a scientific paper, you should quote scientific data and not newspapers. I think the possible effects of the COVID-19 pandemic on ED should be described here better. Previous studies have reported specific vulnerability factors linked to the exacerbation of the ED symptoms in patients (https://doi.org/10.1007/s40519-020-01097-x), as well as discussed the incidents (https://doi.org/10.1192/bjp.2021.105), also with longitudinal evaluations (https://doi.org/10.1016/j.eatbeh.2021.101564)

AR: We totally agree with this comment: quotes 11 and 12 are not meant to be trustworthy sources, instead they were cited to highlight the lack of official data. As it can be seen in the amended version, we have clarified that quotes 11 and 12 (now 15 and 16) were newspaper articles along with the purpose of being cited in the first place: from line 74 “...Official data was not retrieved and information was only spread anecdotally by news outlets (that reported a 30% rise in new cases [15,16]).”

As for the vulnerability factors linked to the exacerbation of the ED symptoms: we further described them in the following lines.

The COVID-19 pandemic and the related restrictions to social life, enacted from march 2020 and persisting in varying degrees until today, heavily impacted individuals, especially the vulnerable ones. Indeed, previous literature reported a significantly poorer mental health status in those who suffered from EDs than the general population during the pandemic [9]. Social isolation, anxiety around family matters and economic factors [9], as well as a perceived low quality of therapeutic relationship and fear of contagion all negatively acted upon their mental well-being [10]. Seeking and receiving the adequate treatments was tougher as a consequence of the reallocation of resources and the disruption of routinary healthcare [11–13].”

- It is not clear to me if second or third admission might be detected and excluded from the analysis.

AR: Second or third admissions are included in the analysis, it was made explicit from line 131: “the investigation focused on medical charts for patients who, coherently with the diagnostic criteria, had been hospitalized for an ED, independently of whether it represented an initial hospitalization or a re-admission.”. We did not find it necessary to carry out separate statistical analyzes between first admissions and re-hospitalizations because our focus is on hospital admission volume for EDs in general.

- for 2021, a quarter of the data is missing. This is a serious limitation of the study! and should be stated clearly in the limits. Is it possible to delete the same quarter in all the years and evaluate the effects? 

AR: For reasons related to the Ministry of Health, unfortunately, data regarding the period after September 2021 are still not available and can’t be updated. We made it more explicit in the limitation of the study. We thank the reviewer for having raised the issue as there was a typo in the previous manuscript version (we wrote 2020 instead of 2021) that was now corrected.

Furthermore, the availability of data was restricted to september 2021.”

During the statistical analysis, it was taken into account the exclusion of the same period of time in each year, but, seeing at the result, we’ve noticed that the increase of EDs’ hospitalization rates is evident and statistically significant even if we consider only ¾ of the last year (2021). We’ve also thought to attempt a statistical interpolation, but it would have created an unnecessary artifact. Based on these argumentations, we think that this underestimation does not affect the results of our study.

-  line 166: why you did not report data for BN?

AR: The proposed adjustments have been included: we added the percentage of each diagnosis out of the total number of EDs’ hospitalizations. We do not consider it was necessary to report the absolute number for each diagnosis as the purpose was that of focusing on hospital admission volume for EDs on the whole.

- lines 187-191: please be aware that your data is in line with other studies showing an increase in pediatric admission for EDs from the end of 2019, especially with the COVID pandemic (see http://dx.doi.org/10.1136/archdischild-2020-319868). Please, revise the paragraph. 

AR: Actually, in the paragraph you mentioned we refer to all hospitalization cases and not only to ED’s hospitalization.

- line 208: it is not clear if the authors are referring to EDNOS or to AN as the most prevalent ED. In any case, it is not true; BED diagnosis has a higher prevalence over the other diagnosis in the general population (see 10.1097/yco.0000000000000449). Maybe, it is true that AN is the most common diagnosis for hospitalization, and it is due to its complication. Moreover, quotes #19 is uncorrected

AR: We agree with your consideration: we’ve now made clear that we were referring not to the general population but to the ED hospitalized population, in which AN is the most diagnosticated.

“As for EDs’ diagnosis trend in hospitalized population, the findings of this study showed that anorexia nervosa was still the most common diagnosis, followed by unspecified eating disorders.”

- Overall, the discussion is too speculative. The authors should be more closed to their data while they discussed the Italian national health system, the definition of onset for ED, the role of pediatricians and the shame felt by parents. These are all aspects far from your data and far from what you can say with your data (and quotes are scarce). 

AR: We know that many arguments in our discussion are speculative. At the same time, at the beginning of the discussion, we tried to provide maximum clarity. We believe it is important to look at the data and to go beyond, visualizing the possible underlying causes behind the observations, in order to evaluate the available facts in a broader perspective. We hope the reviewer will not be disappointed by this initiative.

- please include in your limits the retrospective nature of the study, as well as that your data are limited to the public hospital (I think), and maybe you lost patients that were directly hospitalized in private facilities for ED treatments (maybe this could be more effective for waves 2018-2019 when EDs were less treated in general hospital?).

AR: The proposed adjustments have been included in the limits.

This study has some limitations. Firstly, it is a retrospective study. Secondly, data collected was limited to one age range (5-19), as the focus was on the pediatric and adolescent population; for this reason, it was not possible to assess the entire trend of EDs considering other possible age ranges. Furthermore, the availability of data was restricted to eptember 2021. It is likely that the trend we evaluated continued to evolve during the pandemic, therefore further studies are necessary. Lastly, the collected data were extracted from hospital discharge forms: these documents are operator-dependent, so that their precision or accuracy is by no means certain, even more so concerning a highly specialized topic.

About the data: they comprehend data from both public and private hospitals. Indeed in Italy also private clinics are in network with the Public Health Service. Their data about hospital admission are included in the Italian National Information Discharge System database.

- I think the paper needs a revision by a Native speaker. Some paragraphs are quite hard to read.

AR: As requested, the manuscript has been reviewed by a professional native speaker. Several adjustments were included to reduce faults and increase readability.

Round 2

Reviewer 3 Report

I think the authors have addressed all my concerns.

Author Response

We would like to thank the editor for having allowed this further specification.
Please note the manuscript was amended to include data on first-time admission to better detail the increasing trend observed.
Amendments were added in the Results section, in table 1 foot, and, as requested in the study limitations.
Best Regards.